# Qualitative exploration of health professionals' experiences of communicating positive newborn bloodspot screening results for nine conditions in England

Jane Chudleigh [iD],[1] Holly Chinnery,[2] Jim R Bonham,[3] Ellinor Olander,[4] Louise Moody,[5] Alan Simpson,[6] Stephen Morris [iD],[7] Fiona Ulph,[8] Mandy Bryon,[9] Kevin Southern[10]

For numbered affiliations see end of article.

**Correspondence to**
Dr Jane Chudleigh;
j.chudleigh@city.ac.uk

## ABSTRACT

**Objective** To explore health professionals' experiences of communicating positive newborn bloodspot screening (NBS) results, highlight differences, share good practice and make recommendations for future research.

**Design** Qualitative exploratory design was employed using semi-structured interviews

**Setting** Three National Health Service provider organisations in England

**Participants** Seventeen health professionals involved in communicating positive newborn bloodspot screening results to parents for all nine conditions currently included in the newborn bloodspot screening programme in England.

**Results** Findings indicated variation in approaches to communicating positive newborn bloodspot screening results to parents, largely influenced by resources available and the lack of clear guidance. Health professionals emphasised the importance of communicating results to families in a way that is sensitive to their needs. However, many challenges hindered communication including logistical considerations; difficulty contacting the family and other health professionals; language barriers; parental reactions; resource considerations; lack of training; and insufficient time.

**Conclusion** Health professionals invest a lot of time and energy trying to ensure communication of positive newborn bloodspot screening results to families is done well. However, there continues to be great variation in the way these results are communicated to parents and this is largely influenced by resources available but also the lack of concrete guidance. How best to support health professionals undertaking this challenging and emotive task requires further exploration. We recommend evaluation of a more cohesive approach that meets the needs of parents and staff while being sensitive to the subtleties of each condition.

**Trial registration number** ISRCTN15330120

### Strengths and limitations of this study

► Health professionals involved in the present study were employed in three different National Health Service organisations, increasing transferability of the findings and were very experienced, supporting families for on average 10 years.

► Participants were recruited via email; those with a pre-existing interest in this topic may have been more likely to self-select into the study. These people may communicate results differently than providers who did not participate in the study.

► Health professionals were recruited from clinical teams involved in managing all nine conditions currently included in the newborn bloodspot screening programme in England; previous work has mainly focussed on cystic fibrosis and sickle cell disease.

► The study design, data collection and analysis were influenced by members of the patient and public involvement advisory group and relevant charities.

affected by genetic or congenital conditions;[1] early diagnosis leads to better health outcomes for the child.[1 2] In January 2015, NBS in England and Wales expanded and now covers nine conditions: sickle cell disease (SCD); cystic fibrosis (CF); phenylketonuria (PKU); homocystinuria (HCU); glutaric aciduria type 1 (GA1) medium-chain acyl-CoA dehydrogenase deficiency (MCADD), maple syrup urine disease (MSUD), isovaleric aciduria (IVA) and congenital hypothyroidism (CHT).

In the UK, parents are given information prior to and after birth and are asked to provide informed consent for their baby to take part in the NBS programme. NBS involves a small amount of blood being taken from the baby's heel on day 5 of life. Each

## INTRODUCTION

Newborn bloodspot screening (NBS) seeks to identify presymptomatic babies that are



year, around 10 000 parents (the term 'parents' refers to parents, carers or guardians) of babies born in the UK are given a positive NBS result. This happens around 2 to 8 weeks after birth, depending on the condition.[3 4] Parents are informed of the NBS result, normally on the same day it becomes available, and are asked to bring their baby to a particular hospital for diagnostic testing within a given timeframe depending on the suspected condition.

National Guidance regarding the content and mode of communication between health professionals (HPs) and parents is generic and vague.[1 5] Communication of positive NBS results is often delivered by specialist nurses and is a subtle and skilful task which demands thought, preparation and evidence to minimise potentially harmful negative sequela.[6–10] However, training for this role is challenging due to the fact that for many of conditions, a positive result occurs relatively infrequently and may be communicated by several members of the appropriate team of people.

There is evidence of variations both nationally and internationally with regard to the approaches used to communicate positive NBS results particularly in relation to CF and SCD.[8 10–12] These variations include the different approaches used to communicate the results (such as in person during a home visit or by letter),[11 13] the choice of person or profession to deliver the result and the content of the communication.[14–18]

The findings of Kai *et al*'s study[13] informed the development of the current national guidelines for the communication process in the NBS programme,[1 5] which recommend face-to-face communication by an appropriately trained HP. Despite these guidelines, a study reporting the findings from 67 interviews with parents about their experience of receiving CF or SCD carrier results following NBS indicated that disparity continues to exist regarding how the guidelines are implemented in practice.[7] The findings also indicated variability in the content and the way the result was communicated which led to increased parental anxiety and distress; the perceived lack of knowledge of the person communicating the result was potentially more distressing than the positive finding.[7] This highlights the importance of understanding varied practice in order to identify, share and recommend good practice.

The purpose of this work was to explore HPs' experiences of communicating positive NBS results to highlight differences, share good practice and make recommendations for future research. Experiences of parents are reported elsewhere.[19]

## METHODS

A qualitative exploratory design was employed using semi-structured interviews as part of a large ongoing programme of work.[20]

### Setting

Study sites consisted of three National Health Service provider organisations in England that process comparable numbers of positive NBS reports annually for each of the nine conditions currently included in the NBS programme. These consisted of two in Greater London (served by one NBS laboratory processing 128 positive NBS results in 2017/2018) and one in the West Midlands (whose NBS laboratory processed 129 positive NBS results in 2017/2018).

### Patient and public involvement

Patient and public involvement (PPI) was instrumental in the design and conduct of this work. Parents of babies who had received a positive NBS screening result for one of the nine screened conditions formed a PPI advisory group who met prior to, during and following data collection. Their suggestions were incorporated into the study design, the data collection tools and the data analysis and presentation. In addition, views of representatives from charities for the screened conditions including Metabolic Support UK, the British Thyroid Foundation, the CF Trust and the Sickle Cell Society were also sought. These groups were able to provide advice on sampling methods, choice of study sites and the analysis and presentation of parental interview responses.

### Inclusion and exclusion criteria

HPs involved in communicating positive NBS results in the previous 6 months were invited to take part in the study. HPs who had not been involved in communicating positive NBS results in the last 6 months or who had personal experience of receiving a positive NBS result were excluded.

### Recruitment and sampling

A two-stage sampling approach was employed where participants were first sampled purposively based on their experience with the phenomena of interest, followed by a second stage of snowball sampling where the first participants suggested others. Members of relevant clinical teams (medical consultants; general paediatricians; nurse specialists; specialist screening nurses) were initially identified through individual Trust websites and contacted via email and invited to participate. If no response was received, a follow-up email was sent after 1 week. Identified HPs were asked if there were any other members of the clinical teams that the research team should contact to ensure views were representative. All potential participants were given the choice to participate or not and were reminded of their right to withdraw from the study at any time. Written informed consent was obtained from all participants ahead of the interview.

### Data collection

Semi-structured, face-to-face interviews comprising closed and open-ended questions were conducted by JC and HC from September 2018 to March 2019, to explore HPs experiences of communicating positive NBS to families.

Interviews took place at the participants' place of work in a location and at a time of their choice. Topics covered included how HPs manage communication, examples of situations when it has gone well or not as well as expected and the possible reasons for this. We also explored how participants felt during the experience and asked for any suggested changes to practice. Interviews were audio recorded and transcribed verbatim.

## Data analysis

Interviews were analysed for themes.[21] An inductive approach to data analysis was used and themes generated using a latent approach[21] to provide a deeper understanding of approaches used to communicate positive NBS results to families. The six phases of thematic analysis described by Braun and Clarke[21] guided data analysis.

Two members of the research team (JC and HC) coded one interview transcript separately. These codes were then compared with inform and align code development[22] and a code book was developed. A further four transcripts were then coded separately by the same two members of the research team using the code book. These separately coded transcripts were then compared; inter-coder reliability was 84%. Following this, the same two members of the research team coded the remainder of the transcripts using the code book. Once this initial coding had been completed, all data for each code were compared with to ensure consistency in coding and to enable the codes to be collapsed into themes. All quotes for each theme were collated to inform theme development. This was an ongoing, iterative process; new codes were developed and the definition of codes refined as analysis progressed.[23]

## Positionality and reflexivity

Members of the study team (JC, JRB, LM, FU, MB, KWS) have been involved in or continue to undertake a variety of roles and activities associated with the NBS programme in the UK. It is acknowledged that this could have led to potential bias during data collection and analysis. However, this was balanced by other members of the research team who had previously had minimal involvement in NBS (HC, EKO, AS, SM). Data collection and analysis was mainly undertaken by JC and HC who fall within both camps. Neither JC nor HC were employed in the same organisations where data collection was undertaken.

## RESULTS

In total, 20 HPs were emailed and invited to participate. Two HPs did not respond to the invitation and one HP did not communicate the initial positive screening result and was therefore ineligible. Therefore, 16 face-to-face interviews were conducted with 17 HPs (two requested to be interviewed together); 8 were from the West Midlands and the remaining 9 were split across the two Greater London Trusts. Participants with experience of all the nine screened conditions were included. Interviews lasted on average 37 min (range 19 to 58 min). The sample consisted of eight medical consultants, one medical registrar, seven nurse specialists/advanced nurse practitioners and one screening nurse. Length of experience with newborn screening ranged from 2 to 38 years (median 8 years)

Five themes were identified: communication between HPs; process of communicating with the family; parent and family-centred care; availability of resources and challenges to effective communication. Illustrative quotations are used to support the themes.

## Communication between HPS

HPs reported a range of communication approaches to ensure sufficient information was available to them prior to communicating with families. This started with the laboratory communicating the result to the relevant clinical team in a variety of ways. These included a letter, normally via email or a telephone call followed by an email or a personal visit from a member of the screening laboratory to the clinical nurse specialist, the screening nurse, the on-call consultant or the named consultant depending on the condition, local resources and agreements.

> So we tend to find out from the newborn screening nurses. So, they'll be alerted by the labs and then they would give us [the physician] a call (SS1P202)

The written initial communication consisted of a pro forma which was often developed locally and may or may not have been accompanied by a copy of the NBS card.

> …generally, we get a pro forma from the screening lab that's slightly different I think depending on the screening lab. (SS2P206)
>
> We actually started asking for the card. (SS2P201&02)

Receiving a copy of the NBS card was viewed favourably as it enabled HPs to check referral information and parental contact details if this were found to be ambiguous in any way.

Often, this would initiate a two pronged approach where HPs would commence gathering additional information about the child and family from health visitors (registered nurses or midwives who have undertaken additional training and work mainly with children from birth to 5 years and their families), midwives and/or primary care physicians before contacting the family.

> Sometimes, it is good to know, the family dynamics, social care issues, etc, from somebody [the health visitor or midwife] who's already involved with the family. (SS1P203)

However, clinicians sometimes found it challenging to make contact with health visitors and/or midwives to gather additional information about the family.

> …quite often we are leaving messages to ask them [the health visitor or midwife] to call us back and

quite often we are receiving those calls back, after we've already visited the family. (SS1P203)

Simultaneously, members of the multi professional team would also be contacted such as the physiotherapist, dietician or pharmacist (depending on the condition) to inform them that the child would be attending the hospital either the same or the following day.

> The CF nurse specialist would let other members of the team know, so the physio and dietician know that there was a new baby, positive screen so that they could be on standby, but not necessarily to see the family. (SS1P201)

Similarly, if the plan was to see the child in a local hospital rather than in a tertiary care centre, similar communication would happen between the specialist centre and local clinical teams.

### Process of communicating with the family

The initial contact with the family was undertaken in a variety of ways by different members of the clinical team. This included, face-to-face contact in the family's home, via telephone, text message or letter from the screening nurse, relevant clinical nurse specialist or medical consultant. Some respondents commented that they felt the person who told the family should not be a member of the specialist clinical team who would go on to care for the child and they felt this may taint any ongoing relationship. Others felt it was important that the person who gave the initial positive NBS result should be part of the child's clinical team to start building familiarity and continuity of care or that there should be someone present who is known to the family.

> So the ideal is that either the midwife or the health visitor can come on the visit so that somebody in the family already knows. (SS1P203)

Views regarding how the initial communication with the family was conducted varied. Some clinicians felt it should be face-to-face as it allowed the clinician to gather information about the family that would help to inform the follow-up visit at the hospital with the clinical team.

> You just don't know what's going on in the home environment and you've, sort of, been and witnessed it for yourself and it just gives you a good insight into the family dynamics or what's going on or what support mechanism are in place. You can't get that over the telephone. (SS1P202)

Others felt that a telephone call would be more appropriate to ensure families were told as soon as possible while some felt that a text message asking the family to call them back had worked really well and that a home visit may be quite intrusive. This suggests that sometimes, the approach to communication may not be steered wholly by the needs of the family but sometimes the experience of the person communicating the result.

> You can't really get rid of anyone in someone's house, can you?…You'd be a bit like, 'Right, go now from my house. Get out of my house'…how scary must it be for somebody to turn up on your doorstep (SS2P201&02)

Regardless of who or how the initial communication took place, all respondents acknowledged the importance of 'getting it right'. Respondents felt it was important the person giving the positive screening result to the family was knowledgeable about the condition they would be discussing with the family. Indeed, many felt this was more important than how the information was delivered and could influence perceptions of ongoing care.

> I think the most important thing, in my opinion, is that the person giving that information, the first time, needs to be someone that can answer some questions. (SS1P207)

> I think that first telephone call and the first time you see them is absolutely critical…I think the family's views of what's going to come next will be completely modified by how it's done and their confidence in the service. (SS2P206)

However, it was also acknowledged that although various guidelines and protocols existed for the laboratory staff when processing the NBS card and for treatment and management once the child had been diagnosed, there was a paucity of guidance regarding communication of the initial positive NBS result.

> We've got all the protocols around timelines and KPIs, etc, but the one bit we don't have anything concrete about is who breaks the news, what level of training or experience they need to have before they do it and what the expectation is of what they should cover in that visit. (SS1P203)

Consequently, the content of the initial communication also varied considerably. Some clinicians spoke about having a template they followed which helped them to ensure they imparted all the information required during what could sometimes be a very emotive interaction. Most agreed that they would try to keep the information about the suspected condition during the initial communication quite brief due to the fact it is a screening result and therefore it would not be appropriate to give too much information about a condition that had not been confirmed. In addition, families were often perceived as being unable to absorb the information very well due to the shock of an unexpected result. Finally, clinicians knew that when families are seen the same or following day, they would receive a lot more information and therefore were reluctant to overload them with information during the first contact.

> So we always have said, 'Screening will be rechecked. The bloods will be redone.' (SS1P203)

We don't usually give them much information, they're [the family] usually really upset. (SS2P201&02)

However, HPs did recognise the importance of signposting families to additional information sources such as charity websites and the National NBS website following the initial communication and gave examples of excellent practice.

We always take out suspected leaflets. They always get a copy of something to read. We take out a map for the hospital…so they know where to go to in that building….We leave them a letter that confirms what the screening result is and what that might mean for the baby. On the top of that letter, it has our mobile number and office number to give them permission to ring us if they were at all worrying. (SS1P203)

### Parent and family-centred care

The importance of having a parent and family-centred approach when communicating the positive NBS result was emphasised by all clinicians and was considered to be an example of excellent practice. HPs spoke about the importance of the content of the initial communication. This was subdivided into a beginning, namely, remembering to congratulate parents on the birth of their child. This was followed by a section focussed on tailoring the quantity and level of information given depending on the parental response, not giving too much information or overwhelming parents, being honest and providing emotional support and reassurance where needed.

So, you just have to judge how much information to give them and how best to support them because if parents are crying and upset about the diagnosis, you obviously have to support them in a different way to other parents. (SS1P205)

Finally, ending with a positive message, giving parents' time and making sure the family know that the baby's mother should be encouraged to bring a support person to the first visit. Also, the importance of treating each family as individuals and acknowledging the enormity of the task they are undertaking.

I think it's easy, particularly if you're tired or something, to appear a bit more routine, run of the mill, whereas, obviously, it's a really big thing. This is their child, and this is an important piece of information, and it matters hugely to them. You have to try and reflect that in how you talk to them, rather than them just being another parent of another sickle baby, which is the danger. (SS3P206)

Some clinicians said they endeavour to find out how much parents already know, for instance, if they have another child with the condition or if they have been searching the Internet before embarking on any explanations. Also, gauging how much families want to know

as well as how much families are actually absorbing and when it is time to stop.

…there are some families that come in that want to know every single detail about everything and you try and be systematic in how you deliver that information. There are other parents that you can tell, although you're trying to tell them things, they are not really taking it on board. (SS1P205)

Finally, HPs recognised the importance of supporting the whole family and therefore not making it the responsibility of the parents to share information or educate other family members.

We're there to support whoever. The home visit follow-up that I do, I say, 'All are welcome, you know, if granny and granddad want to be there, if they've got questions, I'm happy to speak with them.' Very often, grandparents, aunts and uncles are there, you know. I've been to one and people have been there on mass, but that's their family support network and if that's usually there in place, we don't exclude anyone. I'm happy to talk to any family member. (SS1P202)

### Availability of resources

HPs at all levels and for all conditions acknowledged a lack of training and related competencies in terms of breaking bad news to families. This resulted in screening nurses in particular developing their own training programme to address this deficit.

…we've had to develop some competencies….but because it's so unique and there aren't other nurses funded through a lab post, that there aren't competencies around. (SS1P203)

Resources available at different hospitals also influenced communication of the initial positive NBS result. Therefore, even if clinicians felt that offering to do a home visit to deliver the initial positive NBS result would be beneficial, the need to prioritise resources often meant this would not be an option.

…we don't have the capacity [to do home visits]….I think it will be good if we can do that, but we definitely don't have the capacity for that. (SS2P203)

One hospital had screening nurses and part of their role was to deliver the initial positive NBS result to families, usually face-to-face in their home. In the other two hospitals, this resource was not available and therefore the responsibility of delivering the initial positive NBS result to families stayed with the relevant clinical team. Even though HPs who were able to, felt very positive about the ability to offer home visits, these were resource intensive particularly when compared with the option to contact parents via telephone.

Probably around 40 minutes is a quick visit. The longest visit I have been there [excluding travel] is probably about three hours. (SS1P203)

The ability to offer timely follow-up appointments with the clinical team both due to clinical need and also to alleviate parental anxiety was also viewed as important. However, many respondents discussed the availability of resources as a potential barrier to this and this seemed particularly evident for babies with SCD.

It can be a week if we've got a slot, or it can be anything up to 4 to 5 weeks….I think it is a very stressful time and some families find that too long. (SS1P209)

For other conditions, clinical need meant that families were seen almost immediately by the clinical team after they had been given the initial positive NBS result.

So, it's all clinically indicated….the rest of the metabolic conditions will be seen the same day. (SS1P203)

Although ensuring families are seen quickly after being told their child has a positive NBS result was seen as important by the clinical team, this also posed problems on occasions in terms of the financial burden for families needing to travel to the hospital at short notice. Currently, there is no budget to prospectively pay for families' travel expenses to attend clinic appointments which means that families are expected to meet these costs in the first instance. Although it may be possible, in certain circumstances, to apply to have these reimbursed at a later date, this does not help families who struggle to pay these costs up front.

…but we have had families who, it has happened where a family said, 'I just don't think I have the money. I can't afford to come' and saying, 'We'll pay you back' doesn't help and that is difficult. (SS2P206)

Some clinicians also felt that the setting within which the family met the clinical team was important in terms of first impressions.

This room was designed that it hasn't got a computer or a phone, unlike our other consultation rooms. So, it's a quiet space to deliver the news to the family. (SS1P202)

Availability of resources for diagnostic testing was also viewed as potentially problematic.

…when we started, the labs were kitted out, such that they would be able to offer a sweat test and a result the same day, any day….because of cuts…they're much less flexible in terms of what they can offer. (SS1P201)

Therefore, although HPs felt very strongly about offering a parent-centred approach to communicating positive NBS results to parents, availability of training, staff and physical space could act as a barrier.

Other concerns related to practical and resource considerations included adding new conditions to the NBS programme and changes in geographical areas covered by screening laboratories particularly in relation to inherited metabolic diseases and CHT.

…they [babies with CHT] are dealt with by 15 different centres…and I think that causes the labs quite a lot of problems. The labs would much prefer fewer centres. (SS2P207)

### Challenges to effective communication

HPs in the current study were asked about their experience of communicating positive NBS result to families. Several staff commented on the personal and emotional impact of this aspect of their role.

Some can be pretty traumatic and you just feel like you've destroyed their world. (SS1P202)

It's very emotional for me sometimes, very, very emotional…the mother, a new born, holding a new born baby crying…based on the result you gave them…you wish you could change the result for them. (SS3P211)

Despite this, no formal mechanisms were in place to support HPs. However, all reported that they had developed their own support mechanisms within their teams.

…everyone's very used to doing it, so everyone completely gets it and completely understands….you have a bit of a debrief with the consultant who's seeing the family. (SS3P202)

HPs also struggled to make contact with parents for a variety of reasons including, the contact details on the NBS card being illegible, parents having moved or staying with relatives after the NBS sample was taken and parents not answering their phones.

You fill in clearly….not flipping scribbled…the name of the person, a first number, a second number of a partner or someone like that…even though we had a picture of the card, they hadn't filled in any of the details (SS2P201&02)

It's not being able to get in contact with people, because they're not picking up on a random London number…The amount of junk calls we all get nowadays, these people are just not picking up the phone. (SS2P201&02)

Once parents had been contacted, other challenges arose related to parental attributes. A common theme related to communicating with parents who did not speak English as a first language. This was not always apparent until after the clinician had attempted to make contact with the parent. This led to parents not understanding what was being communicated, the seriousness of what was being communicated or simply not engaging in the conversation due to lack of understanding.

If English isn't the first language, I certainly find it much more difficult to reassure, and try to be empathetic because you're worried about just getting the basic understanding across. (SS2P206)

Managing families who do not believe the diagnosis either because their child has no signs or symptoms or due to religious or cultural beliefs was also difficult. Such beliefs may also affect engagement with recommended medication and/or treatments. This was particularly evident for those families with a child with SCD.

They just wouldn't accept that they'd got sickle, because they looked well. (SS1210)

…so you get a few that either don't believe the diagnosis or their cultural beliefs are that God will mean that they don't have to do anything medical that we suggest. (SS1P205)

Clinicians commented that this often led to patients not attending clinical appointments which also posed challenges in terms of monitoring.

Other challenges related directly to the NBS programme. The NBS programme is designed to identify babies with a higher chance of being affected by one of the screened conditions, often before they are symptomatic, so they can start treatment as soon as possible if needed, and therefore improve outcomes. However, the need to inform parents that their baby has a positive NBS result for a potentially life changing and/or life limiting condition when the baby is asymptomatic was challenging.

You come to see the doctor because your child is unwell, whereas, this is the opposite, in that, you've got a child who appears perfectly healthy, and you're telling them that they're unwell. (SS1P207)

## DISCUSSION

This study furthers our understanding of HPs' experiences of communicating positive NBS results to families. It is clear that staff involved in communication of positive NBS results are passionate about making sure that although the message is distressing for parents, it is done well. Variation in communication practices continue to exist and is influenced by many factors including resources available but also the lack of clear guidance. This impacted on the methods used to communicate positive NBS results but also the content of the communication to parents. This is supported by previous research which has been conducted both nationally and internationally[7 10–12] and suggests that further guidance may be needed to ensure a more cohesive approach which meets the needs of parents' and staff while being sensitive to the subtleties of each condition. However, the issue of finite resources and the need to prioritise these also needs careful consideration.

An overarching message from the HPs involved in the present study was the desire to ensure communicating positive NBS result to families is parent and family-centred. Once HPs were aware of a positive NBS result, they spent a great deal of time 'setting the scene' by gathering information from various sources in preparation for speaking with the family and organising the follow-up appointment with the clinical team. Often this important and necessary work would be time consuming and labour intensive in terms of identifying individuals who either obtained the original sample (midwives) or might have additional information about family dynamics (health visitor) and may be hindered by poor completion or lack of information on the NBS card.

Current guidance[1 5] does not explicitly state who is the 'right' person to communicate a positive NBS result to families or what training or qualifications they should have and specific training to undertake this role is not available. While this does allow flexibility in terms of resources, it can lead to disparity in terms of parental experience of receiving the NBS result. There is no specific guidance regarding exactly what information should be shared with parents during the initial communication.[5] While some HPs alluded to using an informal checklist, this was not universal and therefore may not lead to consistency between clinicians. However, clinicians did recognise the importance of the person imparting the result, having adequate condition specific knowledge, this is consistent with previous research.[14 15 18] These are common problems that have been highlighted both nationally and internationally.[12] As these are not dependent on specific healthcare systems per se, the findings of the present study could also be extrapolated to screening programmes in other countries.

Clinicians experienced many challenges which hindered the communication of positive NBS results to families. This often stemmed from inadequate information on the NBS card but also parental reactions which could hinder effective communication. When parents are told the NBS result, their baby is often presymptomatic as this is one of the purposes of the NBS programme which means the result is often unexpected.[10 16 17] This can make it difficult for parents to accept their baby may have an underlying health condition which can impact on treatment adherence and affect attendance at follow-up appointments. In addition, parental religious or cultural beliefs could also impact on parent's acceptance of their baby's suspected condition.[10 24] These results demonstrate the importance of always recording clear contact information for all relevant family members on the NBS card as well information about the language spoken, the need for a translator and any relevant religious or cultural information.

The impact on HPs of communicating of positive NBS results to families has rarely been considered. It has been acknowledged that the emotional management of families could lead to additional stress and anxiety.[25] However, HPs in the present study stated they found communicating positive NBS results to families difficult and emotive yet there were no formal mechanisms in place to support them. Despite this, HPs said they felt well supported by their colleagues. However, given the high levels of stress being reported by nurses and doctors and the reported rates of suicide among these professions[26 27] perhaps more consideration needs

to be given to support staff undertaking such emotionally charged endeavours.

The general principles of communicating results that has emerged from this work could be extrapolated for other conditions where screening is recommended in children as well as breaking bad news in general. This might include conditions that may or may not be life altering/threatening but nevertheless can be distressing for parents. For example, delivering results of newborn hearing screening,[28] findings from the physical examination of newborn babies (at birth and 6 to 8 weeks of age) including congenital cardiac abnormalities, congenital cataracts, cryptorchidism, developmental dislocation of the hip and findings from screening of children's eyes at 4 to 5 years of age. It may also be possible to extrapolate findings from the present study for the delivery of bad news to parents in instances such as children newly diagnosed with cancer or following diagnosis of chronic conditions such as diabetes or epilepsy.

### Strengths and limitations

The current study has numerous strengths. HPs involved in the present study were employed in three different trusts; two in Greater London and one in the West Midlands and were very experienced, supporting families for on average 10 years. Clinicians with less experience may have felt even more strongly about the need for clearer guidance. In addition, HPs were recruited from clinical teams involved in managing all of the conditions currently included in the NBS programme. This increases the transferability of the study findings as previous work has mainly focussed on CF and SCD. In addition, the study design, data collection and analysis were influenced by members of the PPI advisory group and relevant charities.

Participants were recruited via email; those with a pre-existing interest in this topic may have been more likely to self-select into the study. These people may communicate results differently than providers who did not participate in the study which may have been based on the findings.

The researchers are experienced in this field which may have biassed data collection and analysis. Following the interviews, a HP event was held and the initial findings of the interviews were presented to the HPs who had been interviewed in each site. The purpose of this was to ensure that the analysis accurately represented their views. Participants agreed that the analysis was correct and priorities identified were accurate.

### Recommendations for practice

The findings from this study suggest a number of recommendations for practice. For example, development of a competency framework for individuals involved in the process of communicating positive NBS results to families would ensure only HPs who are appropriately prepared to undertake this task.

In addition, development of a standard laboratory form for communicating positive NBS results to clinical teams would ensure that when results are received from several laboratories by one clinical team, the information provided by each is consistent.

The development of condition specific checklists for HPs involved in communicating positive NBS result to families would ensure that vital information is consistently relayed to families and less experienced staff would be supported in terms of the information they need to provide. These could also act as an aide-memoire for HPs as it is known that this can be a very distressing time for parents and so it would help them to remain focussed. In addition, this would ensure that clear contact information for all relevant family members including information about language spoken, translation needs and religious or cultural requirements could also be recorded and would be easily accessible for all member of the child and family's care team.

Guidance regarding reliable sources of further information for parents would also reduce alarm that can be caused by accessing unhelpful content on the Internet immediately after the initial communication of the positive NBS result. This might include the use of specifically designed applications or other forms of 'easy to access' and helpful online information for parents.

Finally, regular clinical supervision and emotional support for all staff engaged in such work, should be encouraged to ensure staff are adequately supported to undertake this challenging task.

### CONCLUSION

HPs invest a lot of time and energy trying to make sure communication of positive NBS results to families is done well. However, there continues to be great variation in the way positive NBS results are communicated to parents and this is largely influenced by resources available but also the lack of concrete guidance. Evidence-based guidance derived from HPs' and parents' perspectives is needed to ensure there is parity in terms of communication practices. These should include, what training should be undertaken prior to undertaking this challenging role and guidelines related to what information should be relayed by whom and when. In addition, adequate support mechanism for HPs should be in place.

**Author affiliations**
[1]School of Health Sciences, City University, London, UK
[2]Faculty of Sports, Health and Applied Science, St Mary's University Twickenham, Twickenham, UK
[3]Division of Pharmacy, Diagnostics and Genetics, Sheffield Children's NHS Foundation Trust, Sheffield, UK
[4]School of Health Sciences, City University London, London, UK
[5]Centre for Arts, Memory and Communities, Coventry University, Coventry, UK
[6]Florence Nightingale Faculty of Nursing, Midwifery & Palliative Care, King's College London, London, UK
[7]Primary Care Unit, University of Cambridge School of Clinical Medicine, Cambridge, UK
[8]Florence Nightingale Faculty of Nursing, Midwifery & Palliative Care, University of Manchester, Manchester, UK
[9]Paediatric Psychology and Play Services, Great Ormond Street Hospital For Children NHS Foundation Trust, London, UK
[10]Paediatrics, Institute of Child Health, Merseyside, UK

**Acknowledgements** We would like to thank the Newborn Screening Laboratory Directors in England for agreeing to act as local principle investigators for this study as well as members of clinical teams who gave their valuable time. We would also like to thank all the parents in the Public and Patient Involvement Advisory Group for this study for their invaluable input.

**Contributors** JC made substantial contributions to the conception and design of the work. She acquired and interpreted the data for the work. She was involved in drafting the work, approved the final version to be published and agrees to be accountable for all aspects of the work in ensuring that questions related to the accuracy or integrity of any part of the work are appropriately investigated and resolved. HC acquired and interpreted the data for the work. She was involved in drafting the work, approved the final version to be published and agrees to be accountable for all aspects of the work in ensuring that questions related to the accuracy or integrity of any part of the work are appropriately investigated and resolved. JRB, AS and SM made substantial contributions to the conception and design of the work. They were involved in drafting the work, approved the final version to be published and agree to be accountable for all aspects of the work in ensuring that questions related to the accuracy or integrity of any part of the work are appropriately investigated and resolved. EO was involved interpreting the data for the work and revising the work critically for important intellectual content, approved the final version to be published and agrees to be accountable for all aspects of the work in ensuring that questions related to the accuracy or integrity of any part of the work are appropriately investigated and resolved. LM, FU, MB and KS made substantial contributions to the conception and design of the work. They were involved in drafting the work, approved the final version to be published and agree to be accountable for all aspects of the work in ensuring that questions related to the accuracy or integrity of any part of the work are appropriately investigated and resolved.

**Funding** This study was supported by the National Institute for Health Research (NIHR) (Health Services and Delivery Research (project reference 16/52/25)). The views expressed are those of the authors and not necessarily those of the NIHR or the Department of Health and Social Care.

**Competing interests** None declared.

**Patient consent for publication** Not required.

**Ethics approval** This study was approved by the London Stanmore ethics committee, reference 17/LO/2102.

**Provenance and peer review** Not commissioned; externally peer reviewed.

**Data availability statement** Data are available upon reasonable request. Data are available upon reasonable request from the corresponding author subject to restrictions to preserve anonymity and personal privacy (JC). These data are not publicly available as they contain information that could compromise research participant privacy/consent. Data will be available beginning 1 year and ending 5 years after publication to researchers who propose a methodologically sound proposal. Proposals should be directed to j.chudleigh@city.ac.uk. To gain access, data requesters will need to sign a data access agreement.

**ORCID iDs**
Jane Chudleigh http://orcid.org/0000-0002-7334-8708
Stephen Morris http://orcid.org/0000-0002-5828-3563

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
