## [Reviewer comments · BMJ Open]

ARTICLE DETAILS

TITLE (PROVISIONAL)	A qualitative exploration of health professionals' experiences of communicating positive newborn bloodspot screening results for nine conditions in England
AUTHORS	Chudleigh, Jane; Chinnery, Holly; Bonham, Jim; Olander, Ellinor; Moody, Louise; Simpson, Alan; Morris, Stephen; Ulph, Fiona; Bryon, Mandy; Southern, Kevin

VERSION 1 – REVIEW

REVIEWER	Elza Cloete University of Auckland New Zealand
REVIEW RETURNED	07-Apr-2020

GENERAL COMMENTS	Thank you for the opportunity to review this manuscript by Chudleigh et al. exploring healthcare professional's experiences of communicating positive newborn screening results. The paper highlights important and often ignored aspects in relation to reporting these test results. Overall, the paper reads well. My only criticism would be that it is very lengthy. The key messages from each theme can perhaps be presented in a more concise way. In addition, a few minor suggestions: 1) In the abstract Newborn Bloodspot Screening should be written in full – currently just abbreviated.2) Page 5, row 42 – “communicated by several members” rather than “a several members”3) Page 5, row 25 – it is stated that parents' experiences are reported elsewhere. There is no reference provided.4) Was there any particular reason for inviting these three Trusts to participate in the study? How do they compare in terms of socioeconomic and ethnic demographic?5) Page 8, row 40 – median rather than range is generally reported with range.6) Page 13, row 36 – the statement is made that “the baby's mum should not attend the first appointment at the hospital on her own.” Given that social circumstances sometimes dictates that it is not possible to bring a support person, the sentence should perhaps be re-framed and simply state that “the baby's mum should be encouraged to bring a support person to the first visit”.
--

REVIEWER	Tara Tancred Liverpool School of Tropical Medicine
REVIEW RETURNED	21-Apr-2020

GENERAL COMMENTS

I thank the authors for a very nice paper on an important subject. It's well-written and has clear messaging for the reader. Some points for your discretion:

Strengths and weaknesses

- I'd generally avoid referring to a strength of a qualitative study as being "generalisability", as they are fundamentally not. However, you might wish to refer to the greater depth in data as a result, and the transferability of findings.
- Not interviewing everyone is not a limitation. A possible limitation is that participants were recruited via email, so those with more pre-existing interest in this topic would be more likely to self-select into the study. These people may communicate results differently than providers who did not participate in the study.
- Please ensure acronyms are introduced before they are used (e.g. CF and SCD)

Introduction

- Lines 16 and 17 on page 5 read a bit oddly "the perceived lack of knowledge of the person communicating the result led to additional distress rather than the actual result per se" should maybe be, "it is unclear whether the lack of knowledge of the person communicating the result was more distressing than the positive finding" or something along those lines.
- Perhaps reference in the text where the parent's experiences are reported if this has been published.

Methods

- Provide the approximate numbers of annual NBS positive results (line 41 pg 5)
- It would be important to label your sampling strategy. Here it reads like purposive/convenience sampling, which, as per limitations above, may result in some bias.
- It would also be good to indicate how long participants had to decide if they wanted to participate (e.g. email was sent and one week later, prospective participants were followed up). Or perhaps there was no follow-up, and you simply waited for participants to contact you with a positive response?
- Where did interviews occur?
- Was any software used for coding? Please indicate if so.
- Would be really important to see mention of positionality and reflexivity, particularly as the study team seems to have a lot of clinicians on it, I'm assuming some of whom have personal experience with this topic. Also, it's important to understand JC and HC's relationship to participants—are any of them colleagues, for example?

Results

- These are nicely communicated. A minor point, Keep the quotations with or without quotation marks (e.g. first quotation on page 9 has them, the second doesn't).
- Page 16 line 41, the semi-colon isn't necessary

Discussion

- A well done discussion. Same comments as above around strengths and limitations.
- Might be good to also highlight the usefulness of these findings in terms of a coherent communication strategy whenever HPs are communicating difficult news to parents/patients. Are there any other examples that might be drawn from to suggest what a better

	communication model might look like within your recommendations? For example, providers communicating positive cancer screening results?
--	--

VERSION 1 – AUTHOR RESPONSE

Reviewer 1:

Overall, the paper reads well. My only criticism would be that it is very lengthy. The key messages from each theme can perhaps be presented in a more concise way.

A few paragraphs have been removed from the results section to hopefully remedy this.

1) In the abstract Newborn Bloodspot Screening should be written in full – currently just abbreviated.

Amended

2) Page 5, row 42 – “communicated by several members” rather than “a several members”

Amended

3) Page 5, row 25 – it is stated that parents’ experiences are reported elsewhere. There is no reference provided.

This is in press, reference added to this effect

4) Was there any particular reason for inviting these three Trusts to participate in the study? How do they compare in terms of socioeconomic and ethnic demographic?

It was actually our PPI advisory group who suggested we use these Trusts due to them processing similar number of positive NBS results annually. Under ‘Setting’ I have amended as follows: “These consisted of two in Greater London (served by one NBS laboratory processing 128 positive NBS results in 2017/18) and one in the West Midlands (whose NBS laboratory processed 129 positive NBS results in 2017/18”.

5) 5) Page 8, row 40 – median rather than range is generally reported with range.

Amended

6) 6) Page 13, row 36 – the statement is made that “the baby’s mum should not attend the first appointment at the hospital on her own.” Given that social circumstances sometimes dictates that it is not possible to bring a support person, the sentence should perhaps be re-framed and simply state that “the baby’s mum should be encouraged to bring a support person to the first visit”.

Amended

Reviewer 2:

Strengths and weaknesses

- I’d generally avoid referring to a strength of a qualitative study as being “generalisability”, as they are fundamentally not. However, you might wish to refer to the greater depth in data as a result, and the transferability of findings.

Amended

- Not interviewing everyone is not a limitation. A possible limitation is that participants were recruited via email, so those with more pre-existing interest in this topic would be more likely to self-select into the study. These people may communicate results differently than providers who did not participate in the study.

Amended

- Please ensure acronyms are introduced before they are used (e.g. CF and SCD)

Amended

Introduction

- Lines 16 and 17 on page 5 read a bit oddly “the perceived lack of knowledge of the person communicating the result led to additional distress rather than the actual result per se” should maybe be, “it is unclear whether the lack of knowledge of the person communicating the result was more

distressing than the positive finding” or something along those lines.

Amended

- Perhaps reference in the text where the parent’s experiences are reported if this has been published.

This is in press, reference added to this effect

Methods

- Provide the approximate numbers of annual NBS positive results (line 41 pg 5)

Under ‘Setting’ amended as follows: “These consisted of two in Greater London (served by one NBS laboratory processing 128 positive NBS results in 2017/18) and one in the West Midlands (whose NBS laboratory processed 129 positive NBS results in 2017/18)”.

- It would be important to label your sampling strategy. Here it reads like purposive/convenience sampling, which, as per limitations above, may result in some bias.

Under ‘Recruitment’, the following sentence has been added: “A purposive/convenience sampling approach was employed.”

- It would also be good to indicate how long participants had to decide if they wanted to participate (e.g. email was sent and one week later, prospective participants were followed up). Or perhaps there was no follow-up, and you simply waited for participants to contact you with a positive response?

The following has been added under ‘Recruitment’: “If no response was received, a follow up email was sent after one week.”

- Where did interviews occur?

The following has been added under ‘Data Collection: “Interviews took place at the participants’ place of work in a location and at a time of their choice.”

- Was any software used for coding? Please indicate if so.

No

- Would be really important to see mention of positionality and reflexivity, particularly as the study team seems to have a lot of clinicians on it, I’m assuming some of whom have personal experience with this topic. Also, it’s important to understand JC and HC’s relationship to participants—are any of them colleagues, for example?

The following paragraph has been added to the methods section: “Members of the study team (JC, JRB, LM, FU, MB, KWS) have been involved in or continue to undertake a variety of roles and activities associated with the NBS Programme in the UK. It is acknowledged that this could have led to potential bias during data collection and analysis. However, this was balanced by other members of the research team who had previously had minimal involvement in NBS (HC, EKO, AS, SM). Data collection and analysis was mainly undertaken by JC and HC who fall within both camps. Neither JC nor HC were employed in the same organisations where data collection was undertaken.”

Results

- These are nicely communicated. A minor point, Keep the quotations with or without quotation marks (e.g. first quotation on page 9 has them, the second doesn’t).

Amended

- Page 16 line 41, the semi-colon isn’t necessary

This has been deleted

Discussion

- A well done discussion. Same comments as above around strengths and limitations.

Amended as per above

- Might be good to also highlight the usefulness of these findings in terms of a coherent communication strategy whenever HPs are communicating difficult news to parents/patients. Are there any other examples that might be drawn from to suggest what a better communication model might look like within your recommendations? For example, providers communicating positive cancer

screening results?

The following paragraph has been added at the end of the discussion: “The general principles of communicating results that has emerged from this work could be extrapolated for other conditions where screening is recommended in children as well as breaking bad news in general. This might include conditions that may or may not be life altering/threatening but nevertheless can be distressing for parents. For example, delivering results of newborn hearing screening²⁸, findings from the physical examination of newborn babies (at birth and 6-8 weeks of age) including congenital cardiac abnormalities, congenital cataracts, cryptorchidism, developmental dislocation of the hip and findings from screening of children’s eyes at 4-5 years of age. It may also be possible to extrapolate findings from the present study for the delivery of bad news to parents in instances such as children newly diagnosed with cancer or following diagnosis of chronic conditions such as diabetes or epilepsy.”

VERSION 2 – REVIEW

REVIEWER	Elza Cloete University of Auckland New Zealand
REVIEW RETURNED	05-May-2020

GENERAL COMMENTS	All my queries have been addressed.
-------------------------------------

REVIEWER	Tara Tancred Liverpool School of Tropical Medicine, UK
REVIEW RETURNED	22-May-2020

GENERAL COMMENTS	Many thanks to the authors for submitting this revised manuscript. I think it reads very clearly. I have a few further minor comments and edits, which are mostly typographical. Abstract - Wording of the results is a bit confusing. Should it maybe be “However, many challenges hindered communication including: logistical considerations; difficulty contacting the family and other health professionals; language barriers; parental reactions; resource considerations; lack of training; and insufficient time.” Strengths and limitations - Add a comma in line 7 after “findings”. Introduction - Line 42 on page 4: “b” should be “by”, and line 43 maybe should be “appropriate team of people”? - Line 15 and 16 on page 5 are a different font. Methods - Recruitment should maybe be titled “Recruitment and sampling”, as what’s described is both. - I know I commented on this previously, but on second reading here, the sampling approach described isn’t quite purposive/convenience sampling. Maybe more a two-stage sampling where participants were first sampled purposively based on their experience with the phenomena of interest, then a second stage of snowball sampling where the first participants suggested others. Results
---

	- In lines 8–12 on page 8, the list should be introduced with a colon, and the items separated by semi-colons. Discussion - Line 9 on page 18 should likely be communication “practices”. - Based on what’s described for the PPI, it’s not clear how results were fed back to this group and if/how they influenced analysis, so this might need to come out more clearly in the methods if it’s being suggested as a strength.
--	--

VERSION 2 – AUTHOR RESPONSE

Thank you for forwarding on the reviewers comments. We are very happy to revise the manuscript as per the comments from reviewer 2:

Abstract

- Wording of the results is a bit confusing. Should it maybe be “However, many challenges hindered communication including: logistical considerations; difficulty contacting the family and other health professionals; language barriers; parental reactions; resource considerations; lack of training; and insufficient time.”

This has been changed, thank you for your helpful suggestion

Strengths and limitations

- Add a comma in line 7 after “findings”.

Comma added

Introduction

- Line 42 on page 4: “b” should be “by”, and line 43 maybe should be “appropriate team of people”?

- Line 15 and 16 on page 5 are a different font.

Thank you, all changes made

Methods

- Recruitment should maybe be titled “Recruitment and sampling”, as what’s described is both.

- I know I commented on this previously, but on second reading here, the sampling approach described isn’t quite purposive/convenience sampling. Maybe more a two-stage sampling where participants were first sampled purposively based on their experience with the phenomena of interest, then a second stage of snowball sampling where the first participants suggested others.

This has been added, thank you

Results

- In lines 8–12 on page 8, the list should be introduced with a colon, and the items separated by semi-colons.

This has been changed

Discussion

- Line 9 on page 18 should likely be communication “practices”.

- Based on what’s described for the PPI, it’s not clear how results were fed back to this group and if/how they influenced analysis, so this might need to come out more clearly in the methods if it’s being suggested as a strength.

These have been addressed.